# Research on the Mechanism and Material Basis of Corn (*Zea mays* L.) Waste Regulating Dyslipidemia

**DOI:** 10.3390/ph17070868

**Published:** 2024-07-02

**Authors:** Xiaodong Wang, Lewei Cao, Jiajun Tang, Jiagang Deng, Erwei Hao, Gang Bai, Pei Ling Tang, Jieyi Yang, Huaying Li, Lihao Yao, Cuiwei He, Xiaotao Hou

**Affiliations:** 1Guangxi Key Laboratory of Efficacy Study on Chinese Materia Medica, Guangxi University of Chinese Medicine, Nanning 530011, China; 13104147715@163.com (X.W.);; 2Faculty of Pharmacy, Guangxi University of Chinese Medicine, Nanning 530011, China; 3Guangxi Collaborative Innovation Center of Study on Functional Ingredients of Agricultural Residues, Guangxi University of Chinese Medicine, Nanning 530011, China; 4Guangxi Key Laboratory of TCM Formulas Theory and Transformation for Damp Diseases, Guangxi University of Chinese Medicine, Nanning 530011, China; 5State Key Laboratory of Medicinal Chemical Biology, Nankai University, Tianjin 300071, China; 6Department of Bioscience, Faculty of Applied Sciences, Tunku Abdul Rahman University of Management and Technology, Kuala Lumpur 50250, Malaysia

**Keywords:** *Zea mays* L., corn wastes, chemical composition, dyslipidemia, hyperlipidemia, mechanism study

## Abstract

Corn (*Zea mays* L.) is an essential gramineous food crop. Traditionally, corn wastes have primarily been used in feed, harmless processing, and industrial applications. Except for corn silk, these wastes have had limited medicinal uses. However, in recent years, scholars have increasingly studied the medicinal value of corn wastes, including corn silk, bracts, husks, stalks, leaves, and cobs. Hyperlipidemia, characterized by abnormal lipid and/or lipoprotein levels in the blood, is the most common form of dyslipidemia today. It is a significant risk factor for atherosclerosis and can lead to cardiovascular and cerebrovascular diseases if severe. According to the authors’ literature survey, corn wastes play a promising role in regulating glucose and lipid metabolism. This article reviews the mechanisms and material basis of six different corn wastes in regulating dyslipidemia, aiming to provide a foundation for the research and development of these substances.

## 1. Introduction

Gramineous crops, including corn (*Zea mays* L.), represent one of the three primary agricultural products globally. Along with wheat and rice, these crops constitute roughly 80% of global food production. However, corn production generates a substantial amount of waste. “Crop waste” refers to the parts of crops sown and cultivated by farmers that are not commercially valuable and are not typically used as medicinal or other high-value products in conventional farming practices [1]. Investigating the medicinal benefits of these wastes not only helps address waste accumulation issues but also enhances the potential uses [2]. This approach is crucial for the sustainable advancement of traditional Chinese medicine (phytomedicine) materials.

Hyperlipidemia is a condition affecting lipid metabolism, characterized by elevated levels of plasma total cholesterol (TC), triglycerides (TG), and low-density lipoprotein cholesterol (LDL-C), along with decreased high-density lipoprotein cholesterol (HDL-C) [3]. The prevalence of cardiovascular disease (CVD) is increasing worldwide. In China, for instance, the recently released “China Cardiovascular Health and Disease Report 2022” [4], indicates that CVD prevalence continues to rise, with an estimated 330 million cases linked to risk factors such as tobacco and alcohol consumption, dietary patterns, inadequate physical activity, overweight and obesity, hypertension, and diabetes [5]. Of these risk factors, overweight and obesity are the most prominent. Persistent exposure to these risk factors can lead to dyslipidemia, including hyperlipidemia, which can progress to atherosclerosis (AS). This, in turn, significantly increases the risk of cardiovascular and cerebrovascular diseases such as coronary heart disease and stroke. The devastating consequences of CVD on human health, including the prevalence rate of different dyslipidemias and the distribution of various CVD types, are depicted in Figure 1. Given its significant contribution to CVD, the effective prevention and treatment of dyslipidemia is a top priority in ongoing medical research efforts [6].

The treatment of hyperlipidemia involves dietary control, exercise, and medication therapy. However, long-term use of modern synthetic lipid-lowering drugs (including statins and fibrates) can be associated with side effects and contraindications, such as muscle pain and muscle damage, liver function abnormalities, and digestive issues like diarrhea and nausea, etc. Therefore, the development and application of natural lipid-lowering agents for preventing and treating hyperlipidemia and its complications has become a pressing need.

In recent years, studies have found that corn waste has various medicinal properties [7], e.g., corn bracts can reduce blood lipids [8]; corn silk has the effect of lowering blood lipids [9] and lowering blood pressure [10]; corn straw has antioxidant and hypoglycemic effects [11]; corn cob has the pharmacological effects of lowering blood sugar and regulating blood lipids [12]. One point that cannot be ignored is that the literature research has found that corn waste has significant effects on regulating blood lipids, with various mechanisms [13]. According to traditional Chinese medicine, the mechanisms of lowering blood lipids can be classified as (1) exerting or improving oxidative stress capacity [14], (2) modulating lipid metabolism by inhibiting the absorption and intestinal competition for exogenous lipids, inhibiting endogenous lipid synthesis [15], and promoting intestinal lipid excretion [16], (3) regulating bile acid metabolism by promoting the secretion and efflux of bile acids [17], (4) improving cholesterol metabolism by increasing HDL-C activity and quantity [18], and (5) modulating glucose metabolism to indirectly influence blood lipids by regulating insulin levels and reducing insulin resistance [19] and maintaining gut health by regulating intestinal flora homeostasis to regulate blood lipids [20]. The bioactive functions of corn waste in regulating blood lipids may be related to flavonoids [21], polyphenols [22], polysaccharides [12,23], dietary fiber, oils and fats [24], phytosterols [25,26], and anthocyanins [27]. Thus, by exploring the blood lipid-regulating effects and mechanisms of corn waste, this article contributes to the ongoing search for sustainable and effective approaches to manage dyslipidemia, a major global health concern. 

## 2. Corn Waste and Its Mechanism of Regulating Blood Lipids

Figure 2 illustrates the different components of corn waste, including husks, silk, stalks, leaves, cobs, and bracts. Table 1 shows the phytochemical composition of various corn wastes in regulating dyslipidemia.

The chemical composition and mechanisms of how each component lowers blood lipids are discussed in the following subsections.

### 2.1. Corn Bract

Corn bracts, also known as corn husks or corn huskers, are the outer coating of corn ears. Traditional people often utilize this component for various purposes, including organic fertilizer, livestock feed, fuel, and even waste burning. In addition to these traditional uses, research suggests a promising use of corn bracts in Korean folk remedies for treating diseases such as inflammation and obesity [36]. Additionally, the uses of corn bracts in treating inflammation have also been reported in other countries [37]. Moreover, various tribes in Nigeria have used a decoction made from corn husks to treat malaria [38].

#### 2.1.1. Chemical Composition

Studying the flavonoids and phenolic acids rich in the ethanol extract of corn bract leaves, researchers identified eight structures using modern spectroscopic methods. These structures included four flavonoids (tricin, tricin-5-O-β-D-glucopyranoside, tricin-7-O-β-D-glucopyranoside, and tricin-7-O-(β-D-apiofuranosyl-(1→2))-glucopyranoside IV) and four phenolic acids (chlorogenic acid, ferulic acid, p-coumaric acid, and p-hydroxybenzoic acid) [22,29]. Of these compounds, compounds (**1**) to (**3**) were isolated from corn bracts for the first time. Chemical structures of these compounds are shown in Figure 3. Among the phenolic acids, ferulic acid was identified as the main compound, accounting for 68.37% of the total phenols, followed by p-coumaric acid, p-hydroxybenzoic acid, and chlorogenic acid in lower concentrations [22]. These compounds may regulate blood lipid levels through various mechanisms, such as enhancing fat hydrolysis, boosting HDL-C activity and levels, augmenting antioxidant capabilities, curtailing the creation of natural lipids, and regulating insulin levels to reduce insulin resistance.

#### 2.1.2. The Mechanism of Action

##### Increase HDL-C Activity and Quantity

HDL, the only plasma lipoprotein with an anti-atherosclerotic (AS) effect, promotes and regulates reverse cholesterol transport (RCT), removing excess cholesterol from cells, transporting it to the liver for excretion in bile. This protective function against cardiovascular disease makes HDL levels closely linked to the development and prognosis of coronary atherosclerotic heart disease. The pathogenesis of AS is complex. Studies have shown that the formation of atherosclerotic plaques is caused by circulating factors like oxidized LDL and inflammatory cytokines, as well as various cells in the blood vessel wall. These cells include endothelial cells (VECs), lymphocytes, monocytes/macrophages, and vascular smooth muscle cells (VSMCs) interacting with each other [39].

For vascular endothelial cells (VECs), HDL can stimulate them to synthesize prostacyclin, a potent inhibitor of platelet aggregation and a vasodilator. HDL can also bind to prostacyclin, extending its half-life in the body. Additionally, HDL can reduce DNA synthesis in VSMCs induced by epidermal growth factor, potentially contributing to its anti-thrombotic effects. HDL regulates endothelial function by stimulating endothelial cells to produce nitric oxide (NO) and relax blood vessels [40]. The aqueous extract of corn bracts has been shown to increase serum levels of both endothelin and prostacyclin in the VEC of atherosclerotic rabbits [41]. Prostacyclin, as mentioned earlier, is a potent platelet aggregation inhibitor and a vasodilator. Its metabolites can also reduce cholesterol deposition within cells by increasing cholesterol ester catabolism [42]. Normal VEC function relies on a balance between the vasodilator factor NO and the vasoconstrictor factor endothelin (ET). These factors serve as important indicators for evaluating VEC health [43]. Studies have shown that corn bract water extract can help regulate high-fat metabolism in mice. This may be due, in part, to its ability to modulate serum levels of NO and ET, potentially reducing endothelial apoptosis (cell death) and promoting repair, thereby enhancing resistance to high-fat diet-induced damage [44].

Secondly, corn bracts can control the excessive proliferation of VSMCs, thereby contributing to the regulation of AS. The mechanism is linked to HDL inhibiting the oxidation of LDL [40]. Excessive proliferation of VSMCs is an essential step in causing AS. Corn bract may help regulate AS by influencing VSMC activity through a decreased proliferation rate and an increased apoptosis rate. Additionally, studies have shown that corn bracts can improve blood lipid profiles in rabbits with hyperlipidemia. This includes reductions in total cholesterol (TC), triglycerides (TG), LDL, and very low-density lipoprotein (VLDL), while also increasing HDL to a certain extent and improving the degree of aortic intimal atherosclerotic plaque [45]. These findings suggest that corn bracts may play a crucial role in regulating lipid metabolism in vivo, potentially helping to prevent disorders caused by a high-fat diet (HFD). Furthermore, corn bracts have also been found to regulate genes and proteins related to the apoptosis of VSMCs, such as p53, Fas, Bcl-2, and caspase-3 in AS animals [46]. Flow cytometry experiments have shown a decrease in the expression of genes and proteins related to apoptosis, and a decrease in the apoptosis rate of VSMCs, after the administration of corn bract decoction [47,48,49]. This suggests that corn bracts may promote VSMC apoptosis, potentially contributing to the control of excessive VSMC proliferation in the early stages of AS [48]. Studies have shown that, when genes involved in apoptosis are not functioning correctly in AS, vascular wall cells fail to undergo programmed cell death at a normal rate. This leads to an accumulation of cells and thickening of the intima of the blood vessel, ultimately narrowing the lumen [50]. This narrowing of the lumen significantly promotes the progression of plaques, especially in the later stages of the disease [50]. Bcl-2 is a protein that inhibits cell death. In AS, VSMCs can take up oxidized-LDL to form foam cells [51]. These foam cells, however, express high levels of Bcl-2, which can inhibit their own apoptosis. This creates a vicious cycle, where the cells fail to die and continue to the formation of a necrotic core within atherosclerotic plaques [52]. Based on these findings, it is speculated that corn bracts, similar to many other Chinese medicinal materials, may exert their anti-atherosclerotic effects through several mechanisms, including increasing HDL levels and potentially inhibiting the formation of oxidized LDL and foam cells [53].

In addition to its previously mentioned effects, HDL can also inhibit the expression of adhesion molecules on endothelial cells. This reduces the recruitment of blood monocytes to the arterial wall, further contributing to the corn bract role in treating AS. Experiments have proven that corn bract decoction can reduce the expression of CD44 [54]. CD44 is a molecule found on various white blood cells and plays a diverse role in cellular functions. Recent research suggests a link between CD44 expression and cell apoptosis. Studies have shown that mice lacking both apoE and CD44 genes exhibit reduced inflammatory cell and vascular cell activation [55]. Furthermore, CD44 promotes the adhesion of activated lymphocytes to endothelial and smooth muscle cells, facilitating their migration to sites of arteriosclerotic damage [55]. Interestingly, experiments using a rabbit model with high-fat-diet-induced AS have shown that corn bract decoction can reduce the expression of CD44 in the leukocytes [54]. This finding suggests a correlation between decreased CD44 expression and the protective effects of corn bracts against AS. It highlights the potential of corn bract decoction in regulating CD44 expression, thereby contributing to its anti-atherosclerotic effects [54].

##### Regulate Insulin Levels to Reduce Insulin Resistance and Promote Lipolysis of Adipose Tissue

Insulin resistance inhibits the breakdown of adipose tissue, stimulates the breakdown of TG-rich lipoproteins in the bloodstream, and activates HMGCR activity to promote cholesterol synthesis in the liver. This leads to the delayed clearance of TG from the blood, contributing to hypertriglyceridemia [56]. Therefore, managing insulin resistance is also crucial for improving dyslipidemia in diabetes.

Both α-amylase and α-glucosidase are enzymes that breakdown carbohydrates in the digestive system. Inhibiting these enzymes can be a strategy for managing diabetes by preventing postprandial hyperglycemia levels in diabetic patients. A study by Muhammad Riaz [57] investigated the antidiabetic potential of a methanol extract from corn bracts. The results showed that the corn bract extract exhibited a significant inhibitory effect on α-amylase, with an inhibitory activity of 62.47%. This inhibitory activity was even higher compared with the reference drug acarbose. The observed inhibitory effect may be attributed to the presence of phenols and flavonoids in the methanol extract, which are known to possess various bioactivities [57].

##### Improve Antioxidant Capacity

Hyperlipidemia can disrupt the body’s natural balance between free radicals and antioxidants [58]. Studies suggest that corn bract leaves can improve this imbalance by protecting VECs from damage and by removing excess oxygen-free radicals from the body [57]. These protective effects might be attributed to the presence of polyphenols in corn bracts. Polyphenols are natural compounds with antioxidant properties, meaning they can help neutralize free radicals and potentially contribute to the overall health benefits of corn bracts [59].

Researchers investigated the effects of corn husk meal (CHM) on the antioxidant response of Nile tilapia exposed to hypoxic conditions. The Nile tilapia were fed with either CHM-supplemented feed (experimental group) or a regular diet (control group). The results proved that CHM supplementation increased the activity of CAT in the liver of Nile tilapia under hypoxic conditions. However, the activity of other antioxidant enzymes—superoxide dismutase (SOD) and glutathione peroxidase (GSH-Px)—remained unaffected. The increase in CAT activity suggests that CHM may enhance the liver’s antioxidant response during hypoxia. This might be due to the presence of phenolic compounds in CHM, which are known to possess antioxidant properties. Increasing the body’s and liver’s antioxidant ability may potentially promote lipid metabolism in the liver, thereby contributing to the management of hyperlipidemia [57]. These findings are aligned with the results reported by Kyung-Ok Shin [36] in their study using mice as the animal model. The study showed that corn husk powder could reduce liver weight and decrease blood TG and leptin levels in mice with high-fat-diet-induced hyperlipidemia. Additionally, the study observed an increase in HDL-C. While the exact mechanisms require further investigation, these results suggest that corn husk powder has antioxidative effects and might regulate blood lipids [36]. Studies have explored the potential material basis for corn bract’s health benefits. Research by Zhao and Kang [60,61] focused on corn bract polyphenols, which are natural compounds with antioxidant properties. The team employed response surface methodology to optimize the extraction of corn bract polyphenols through microwave-assisted extraction. The study found that corn bract polyphenols exhibited stronger antioxidant activity than BHT, a common synthetic antioxidant. The IC_50_ of corn bract polyphenols against hydroxyl and DPPH radicals were reported at 4.28 μg/mL and 5.84 μg/mL, respectively, whereas 23.41 μg/mL and 27.57 μg/mL, respectively, were reported for BHT. A study by Muhammad Riaz [57] found that a methanol extract of corn bract exhibited a high DPPH scavenging activity, reaching 88.53%. The observed antioxidant effects might be attributed to the presence of flavonoids and phenolic substances in corn bract extracts. Studies have shown that tannins, flavonoids, and polyphenols can help remove excess lipids [22,62,63]. Some studies have compared the antioxidant activity of corn bract extracts with established antioxidants like vitamin C. The results suggest that corn bract extracts exhibit a higher DPPH free radical activity than vitamin C within a specific concentration range [62].

##### Inhibit the Biosynthetic Pathway of TC and TG

Several natural products can lower blood lipid levels by inhibiting the body’s pathway for synthesizing TC and TG [30]. Similarly, corn bract may also exert its lipid-lowering effects through this pathway.

In the studies by Wang [21] and Zhang et al. [29], tricin and its derivatives as the main flavonoids present in the ethanol extract of corn bracts were reported. These findings contribute to the understanding of the bioactive compounds potentially responsible for corn bract’s lipid-lowering activity. Tricin has shown promise in regulating blood lipids. A study by Dabeen Lee [31] investigated its effects in mice fed a high-fat diet. The study found that tricin administration in mice led to significant reductions in serum and liver TG, body fat content, and liver enzyme (ALT and AST) levels. These findings suggest potential benefits for managing obesity and abnormal lipid metabolism. Western blot analysis revealed that the tricin’s anti-obesity effect might be due to its influence on several factors involved in fat production and storage within cells. These factors include enzymes like fatty acid synthase (FAS), stearoyl-CoA desaturase 1(SCD-1), extension of long-chain fatty acid family member 6 (elovl 6), glycerol-3-phosphate acyltransferase (GPAM), and diglycerol ester acyltransferase, and proteins like peroxisome proliferator-activated receptor gamma (PPAR-γ) and CCAAT/enhancer binding protein α (C/EBP-α), which play a role in fat cell development. The study suggests that tricin’s anti-obesity and lipid-lowering effects might be mediated by activating adenosine 5′-monophosphate (AMP)-activated protein kinase (AMPK), a signaling pathway involved in energy metabolism. Activated AMPK can regulate cholesterol and TG synthesis by inhibiting enzymes involved in their production and promoting fatty acid oxidation [32]. Phosphorylated AMPK regulates body lipids through various pathways, including (1) inhibiting the rate-limiting enzyme HMGCR, an enzyme crucial for cholesterol synthesis, (2) inhibiting the expression of sterol regulatory element binding protein 1c (SREBP-1c), a protein that regulates fatty acid synthesis enzymes (FAS and acetyl Co-A carboxylase (ACC)), and (3) directly inhibiting ACC activity, promoting fatty acid oxidation and reducing TG synthesis [32,33]. As shown in Table 2.

### 2.2. Corn Stalks

Corn stalks, also known as corn stover, are composed primarily of the stems, leaves, leaf sheaths, and tassels of the corn plant, excluding the ears. They are often used as livestock feed, industrial materials, or fuel. In traditional Chinese medicine practices, corn stalks are widely used to reduce swelling, and they have diuretic, heat-clearing, and detoxifying properties. Standard application methods involve decoction, stewing, or incorporating them into herbal packs.

In traditional Indian medicine, corn stalks are used to treat various ailments like urinary tract infections, indigestion, and fever. They are typically prepared as a decoction or ground into powder for consumption. In Mexican traditional medicine, corn stalks are used in herbal teas, believed to benefit the kidneys and digestive system. This tea is thought to help alleviate stomach pain and improve digestion. In Nigeria, corn stalks are employed for their perceived antimicrobial and anti-inflammatory properties. They are commonly used to treat mouth ulcers and inflammatory diseases.

#### 2.2.1. Chemical Composition

Corn stalks contain various bioactive chemical substances, including flavonoids, terpenes, phenylpropanoids, and alkaloids. Scholars isolated and identified a pair of diastereoisomers salcolin A and salcolin B from corn stems [80]. The terpenoids isolated from corn stems include zealexin A1 [81], zealexin A2 [81], zealexin A3 [81], zealexin B1 [81], zealexin A4 [67], kauralexin A1 [64], kauralexin A2 [64], kauralexin A3 [64], kauralexin B1 [64], kauralexin B2 [64], and kauralexin B3 [82]. Of these terpenoids, kauralexin showed significant antifeedant activity in Nubian voles. Additionally, some simple phenylpropanoid compounds were also isolated from corn stalks [83], such as (E)-methyl p-coumarate, trans-A methyl coumaric acid, methyl p-coumaric acid, 1,3-O-diferuloylglycerol,tetrahydro-4,6-bis(4-hydroxy-3-methoxyphenyl)-1H,3H-furo [3,4-c]furan-1-one. The alkaloids N-trans-ferroyltryptamine [84], N-(p-coumaryl) serotonin [84], and N-(p-coumaryl) tryptamine [82] in corn stalks are specific inhibitors of acetylcholinesterase.

A study by Yang [11] investigated the hypoglycemic effects of different corn parts. Chemical analysis of corn straw identified four flavonoids, namely trans-4′-methoxy-4-Nitrochalcone, diethyl2-acetamido-6-(1-cyano-2-ethoxy-2-oxoethyl)-1,3-azulenedicarboxylate, 4′-methylepigallocatechin-3′-glucuronide(Complex), and 4′,8-dimethoxy-epigallocatechin-3′-glucuronide. Structural information of these compounds is shown in Figure 3 (Compounds (**9**)–(**12**)). These compounds were reported to link with lowering blood sugar and regulating blood lipids in previous research.

#### 2.2.2. The Mechanism of Action

The corn stalk has been shown to exhibit antioxidant and hypoglycemic activities, potentially affecting lipid metabolism, glucose metabolism, energy metabolism, and other pathways in IR-HepG2 cells, thereby contributing to their hypoglycemic effects [35]. As shown in Table 2.

Yang [11] investigated the potential health benefits of corn stalk and corn silk. The researcher established an in vitro model to assess the inhibitory activity of these materials against α-glucosidase and α-amylase enzymes. The study found that extracts containing total flavonoids from both corn straw and corn silk exhibited antioxidant and hypoglycemic properties. The study employed four methods (DPPH, ABTS+, hydroxyl radical, and FRAP) to compare the antioxidant activity of total flavonoids from various corn parts. The results showed that corn straw flavonoids exhibited strong DPPH, ABTS+, ·OH radicals scavenging ability, and Fe^3+^ reducing power. Researchers then used liquid mass spectrometry to identify potential hypoglycemic active compounds within different corn parts. Additionally, ultrafiltration affinity technology helped screen active ingredients from corn flavonoids that inhibited the α-glucosidase enzyme. The study successfully identified eight promising compounds across various parts of the corn plant, with four found specifically in corn straw. Structural information of these compounds is shown in Figure 3 (Compounds (**9**)–(**12**)). Additionally, cell-based models involving insulin-resistant (IR) human hepatoma cells (HepG2) and high-glucose-damaged human umbilical vein endothelial cells (HUVEC) were established to assess the activity of corn straw and corn silk flavonoids. The researcher then used hydrogen nuclear magnetic resonance spectroscopy (1H NMR) combined with cell metabolomics technology to explore how the flavonoids specifically affected the metabolism of IR-HepG2 cells. The results suggest that total flavonoids from various parts of corn can increase glucose uptake by enhancing the ability of cells to utilize glucose to varying degrees. Additionally, flavonoids also protect HUVEC cells from damage by high sugar levels. Interestingly, the corn waste-derived flavonoids seem to primarily influence pathways related to sugar metabolism, energy metabolism, and lipid metabolism in IR-HepG2 cells. This suggests that corn waste flavonoids might exert hypoglycemic effects and regulate blood lipid levels through these pathways. However, further research is needed to confirm these possibilities [11].

### 2.3. Corn Silk

Corn silks, also known as Stigma maydis, are the long thread-like fibers at the top of an ear of corn. Corn silk has long been used in traditional medicine around the world, including China, Turkey, the United States, and France. Traditionally, corn silk has been used to treat various ailments such as cystitis, edema, kidney stones, diuretics, prostate disorders, urinary infections, bedwetting, and obesity [85], attributed to its diuretic, choleretic, hypoglycemic, and blood pressure-lowering effects [86], as recorded in the *Chinese Pharmacopoeia*. Modern pharmacological research suggests corn silk may have broader health benefits beyond its traditional uses. These potential benefits include diuresis, reducing swelling, calming the liver, promoting choleretics, anticancer, antioxidant, neuroprotective, lowering blood sugar, blood pressure, and blood lipids levels [87].

#### 2.3.1. Chemical Composition

Corn silk contains a variety of chemically active ingredients, including flavonoids, phenols, terpenes, polysaccharides, amino acids, and organic acids, among others [88]. Flavonoids are particularly abundant. Flavonoid glycosides, flavonols, and isoflavones are among the main flavonoids in corn silk. Apigenin, luteolin, robin, and daidol are common parent core structures. A scientific study identified 35 flavonoids in concentrated corn silk extracts using the LC-MS/MS method. Luteolin-C-glycoside and apigenin-C-glycoside accounted for 40% of the total flavonoids found [69]. Another study by Yi Ting [65] used HPLC-Q-TOF-MS to analyze the total flavonoids in corn silk prepared by the reflux extraction method, followed by treatment using macroporous resin. This method successfully identified 19 unique flavonoids (10 flavonoid glycosides and 9 flavonols). The recent phytochemical literature suggests that corn silk contains diverse flavonoids, with 80 being identified, including luteolin, apigenin, and myristin. It is important to note that corn silk’s active ingredients extend beyond just flavonoids. Various glycosides, such as -glycosides and c-glycosides, of flavonoids were also detected [70]. 

Additionally, sterols, which are naturally occurring active substances derived from plants and animals with various uses in medicine, health care, food, and other fields, were also detected. Numerous studies show that corn silk appears to be rich in beta-sitosterol. A study by Tian [89] successfully isolated 17 compounds from corn silk using petroleum ether and ethyl acetate. Of these compounds, stigma-4-ene-3β, 6-β-diol, stigma-4,22-diene-3β, 6-β-diol, stigma-5-ene-3β, 7α-diol, and ergosterol peroxides (the sterols) were detected.

Until now, studies have reported that the extraction of terpenes from corn silk is scarce.

However, the terpenes content is relatively high in corn silk. Zhao Min et al. [66] used combined silica gel and Sephadex LH-20 column chromatography to separate and purify three terpenes, namely 19-hydroxy-r-karene-15-ene-17-carboxylic acid, 17 -hydroxy-r-karene-15-en-19-oleic acid, and 3α-hydroxy-r-karene-15-en-17-oleic acid-19-methylcarboxylic acid, from corn silk. Other than that, amino acids, short-chain organic acids, and long-chain organic acids were also detected in corn silk. The results suggest that corn silk contains 16 types of amino acids. Among them, glutamic acid and aspartic acid have the highest content, followed by the four essential amino acids, namely leucine, phenylalanine, threonine, and valine [90]. In total, 55 organic acids, including linoleic acid, lactic acid, behenic acid, vanillic acid, and stearic acid, were found in corn silk. Additionally, corn silk polysaccharides were also found in the aqueous extracts of corn silk. The polysaccharide content is relatively high in corn silk, reaching 4.87% in the dried form [89]. The main sugar constituents of cornsilk polysaccharides are mannose, lactose, galactose, rhamnose, arabinose, xylose, and glucose [68].

Of the diverse bioactive constituents in the aqueous extract of corn silk [91], flavonoids and polysaccharides have been experimentally proven to exhibit blood lipid regulating effects [68,71].

#### 2.3.2. The Mechanism of Action

Previous studies have proposed that corn silk regulates blood lipids by exerting or improving oxidative stress capacity, inhibiting the absorption of exogenous lipids, inhibiting the synthesis of endogenous lipids, and promoting the decomposition of cholesterol and fatty acids. As shown in Table 2.

##### Promote the Breakdown of Cholesterol and Fatty Acids in the Body

Previous research suggests that corn silk decoction can lower blood lipids, TC, TG, LDL, and free fatty acid levels, reduce weight, Lee’s index, and body fat of hyperlipidemic rats, thereby regulating lipid metabolism disorders [91].

In a metabolomic study, the metabolic pathways of hyperlipidemic rats with severe glucose and lipid metabolism disorders were restored after treatment with corn silk decoction. Corn silk water extract activates the AMPK signaling pathway, promotes the breakdown of cholesterol and fatty acids in the body, and reduces blood lipid levels by increasing the expression of AMPK, rates limiting enzyme CYP7A1 for cholesterol decomposition, and rates limiting enzyme CPT1 for fatty acid decomposition, thereby improving lipid metabolism in hyperlipidemic rats [91].

##### Improve Oxidative Stress Capacity and Enhance Reverse Cholesterol Transport

Corn silk increases free radical scavenging enzyme activity by regulating lipid metabolism and resisting free radicals and peroxidized lipids, and lowering blood sugar, hence protecting the cardiovascular system and preventing AS.

The standard soil theory proposes that oxidative stress is the underlying basis for insulin resistance, diabetes, and CVD [92]. Other than polysaccharides, research has proven that corn silk’s flavonoids can prevent lipid metabolism disorders and abnormal changes in blood rheology indicators caused by a high-fat diet. Based on results from a previous study, corn silk flavonoids significantly reduced serum TC, TG, and LDL-C levels and increased HDL levels (*p* < 0.05) in hyperlipidemic diabetic rats, indicating that corn silk flavonoids may increase the reverse transport of cholesterol by increasing HDL and eliminating excess cholesterol in the peripheral tissue cells, thereby preventing AS [71]. Additionally, the results also suggest that corn silk flavonoids significantly reduced serum and liver MDA levels, increased SOD levels, and were proven to have significant free radical scavenging and antioxidant effects in hyperlipidemic diabetic rats. It is worth highlighting that corn silk flavonoids can increase the activity of free radical scavenging enzymes and resist free radicals and lipid peroxidation in the body by regulating lipid metabolism, thereby lowering blood sugar and lipids, thus protecting the cardiovascular system and preventing AS [71].

In addition, the low molecular weight corn silk polysaccharide CSP-3 is a potential functional food additive with hypolipidemic activity. Its molecular weight and monosaccharide composition information are shown in Table 1. In an in vivo study to assess the CSP-3’s blood lipid-lowering mechanism using the hyperlipidemic mouse model, it was proven that antioxidant balance plays a crucial role. Increasing activity of LPL and HL enzymes ensures a balance of blood lipids by reducing TC, TG, and LDL-C content, while increasing HDL-C content. Meanwhile, the balance of the antioxidant system is achieved by increasing the activity of SOD, GSH-px, and CAT antioxidant enzymes [93].

##### Inhibit Cholesterol Synthesis Pathway

A study by Miura [94] utilized a special diet feeding and Triton-wr 1339 (tetrabutyric phenolic aldehyde) to induce hyperlipidemia in mice, aiming to investigate the therapeutic effect of corn silk aqueous extract. The results showed that the group fed with corn silk aqueous extract exhibited significant reductions in TC and TG levels in the blood of normal mice. Compared with the control group, cholesterol levels in high-cholesterol mice were significantly reduced, and TG levels showed a downward trend. Cholesterol was reduced in mice with tetrabutanol-induced hyperlipidemia, but TG did not change. These results suggest that the cholesterol-lowering effect of corn silk aqueous extract may be due to preventing cholesterol synthesis in the liver [94].

In addition, corn silk extract can also lower blood sugar, and improve blood lipids and blood rheology to varying degrees in diabetic foot model mice induced by a high-fat high-sugar diet and STZ intraperitoneal injection. The study showed that the therapeutic effects on blood sugar, blood lipids, and blood rheology improved with increasing dosage. Specifically, corn silk aqueous extract significantly reduced blood lipid levels. The TC, TG, and LDL were significantly decreased, while HDL levels rebounded (*p* < 0.05) [95]. Further research supports the positive therapeutic effects of corn silk polysaccharides on blood sugar and lipids [96]. A comparative study found corn silk polysaccharide to be more effective than Pu’er tea polysaccharide in lowering TC and TG levels in rats. This suggests its ability to clear intravascular blood lipids and potentially reduce the risk of *AS* and cardiovascular and cerebrovascular diseases. Acid corn silk polysaccharide (ACSP) has also demonstrated promising hypolipidemic activity. Studies showed significant reduction in plasma TC, TG, LDL-C, and Kch values (a market for CVD risk) in the high-dose ACSP group compared with the model group (*p* < 0.05). A smaller Kch value indicated a lower CVD risk. The study showed that the Kch value was significantly reduced (*p* < 0.05) after administration of acidic polysaccharide, proving its promising hypolipidemic activity [68]. However, these findings warrant further investigation to elucidate the underlying mechanisms.

##### Promote the Secretion and Excretion of Bile Acids

Research suggests that corn silk polysaccharide SMP-1 has strong lipid-regulating effects. An infrared spectroscopy technique was used to confirm the structural characteristics of corn silk polysaccharides and found they mainly contained sugars like galactose, glucose, and mannose, along with a high amount of pectic acid. In vivo experiments using mice have shown that corn silk polysaccharides can significantly reduce lipid levels in both their blood and liver. This confirms their ability to lower lipid levels. Notably, SMP-1 can effectively bind to bile acids in vitro. This binding reduces TC, TG, and LDLc levels, while increasing LDLc levels in the blood of hyperlipidemic mice induced by P407. Moreover, pretreatment with SMP-1 (300 mg/kg) can also significantly reduce fat accumulation in the livers of hyperlipidemic mice. This suggests that corn silk polysaccharides may lower lipid levels by interacting with bile acids, affecting how they are reabsorbed and excreted by the body, ultimately influencing fat metabolism [97].

### 2.4. Corn Bran

Corn bran is a by-product of corn starch processing, also known as corn fiber, accounting for about 7–10% of corn. It is commonly used as feed after processing.

#### 2.4.1. Chemical Composition

Corn bran is rich in lignan compounds, dietary fiber, alkaloids, and other chemical substances. While corn stover contains phenylpropanoid components, corn bran boasts a more complex profile of lignan compounds. These compounds are mainly isolated from the bran through various methods like acid hydrolysis, alkali hydrolysis, gel column chromatography, and semi-preparative RP-HPLC. This process has yielded a variety of lignans, including dimers, trimers, tetramers, and glycoside-containing lignans with specific structures, such as 5-O-(trans-feruloyl)-L-arabinofuranoside,O-β-D-xylopyranosyl -(1→2)-[5-O-(trans-feruloyl)-L-arabinofuranoside, O-L-galactopyranosyl-(1→4)-O-D-xylopyranosyl-(1→2)-[5-O-(trans-feruloyl)-L-arabinofuranoside] [98], p-coumaroylated-L-arabinofuranoside, α-Lgalactopyranosyl-(1→2)-β-D-xylopyranosyl-(1→2)-5-O-transferuloyl-L-arabinofuranoside, α-D-galactopyranosyl-(1→3)-α-Lgalactopyranosyl-(1→2)-β-D-xylopyranosyl-(1→2)-5-O-transferuloyl-L-arabinofuranoside, α-D-xylopyranosyl-(1→3)-α-Lgalactopyranosyl-(1→2)-β-D-xylopyranosyl-(1→2)-5-O-transferuloyl-L-arabinofuranoside [99], 8-8(cyclic)-dehydrodiferulic acid [100], 8-O-4,8-O-4-dehydrotriferulic acid, 8-8(cyclic), 8-O-4-dehydrotriferulic acid [100], 8-8(tetrahydrofuran)-dehydrodiferulic acid, 8-5(noncyclic)-dehydrodiferulic acid [73], 8-5(noncyclic)/5-5-dehydrotriferulic acid, 8-8(tetrahydrofuran)/5-5-dehydrotriferulic acid, 4-O-8/5-5/8-O-4-dehydrotetraferulic acid, and 4-O-8/5-5/8-5(noncyclic)-dehydrotetraferulic acid [73]. Corn bran is also a valuable source of dietary fiber. Studies suggest that it can improve gut health by promoting the growth of beneficial intestinal flora. The feruloyl oligosaccharide of corn bran significantly increased the intestinal bacteria richness and diversity of mice compared with the control and xylo-oligosaccharides [74]. Additionally, it also contains two alkaloid components: N,N’-diferuloylputrescine [75] and N-p-coumaroyl-N’-feruloylputrescine [75].

Despite being a waste, corn bran is rich in valuable nutrients, including proteins, essential amino acids, essential fatty acids, ferulic acid, and other phenolics, tocopherols, and β-glucans. These high-value compounds are often used as additives in food and are also consumed in folk practice to reduce blood lipids [76,101].

#### 2.4.2. The Mechanism of Corn Bran in Lowering Blood Lipids

The lipids in corn bran can be extracted with solvents and then purified to obtain functional corn bran oil. Research shows that corn bran oil contains 13.9% phytosterols and 86.1% fatty acids, with sitosterol accounting for more than 80% of the total phytosterols [72]. Experiments have shown that corn bran oil has the function of reducing animal plasma cholesterol. The cholesterol-lowering effect is likely due to the high phytosterol content [25]. Unlike corn oil, which is extracted from the corn germ, corn bran oil boasts a higher phytosterols content, reaching 10–15%. Phytosterols are associated with various health benefits, including preventing CVD, inhibiting tumors, promoting metabolism, and regulating hormone levels [102]. One mechanism by which corn bran oil reduces cholesterol is through its phytosterols content. These plant compounds have a structure similar to cholesterol. Because of this similarity, phytosterols can compete with cholesterol for absorption in the small intestine. This competition helps to reduce the amount of cholesterol being absorbed [103]. Additionally, corn bran oil can also lower plasma cholesterol levels by affecting bile acid production. When cholesterol levels are high, the liver produces more bile acids to eliminate excess cholesterol. This ultimately leads to lower hepatic microsomal cholesterol and alters the hepatic cholesterol metabolism.

A study by Tripurasundari Ramjiganesh [104] found that the addition of corn bran oil to the feed of obese guinea pigs could increase plasma LDL-C levels. However, as the dose of corn bran oil increased, the levels of LDL-C gradually decreased. Compared with the control group, corn bran oil intake significantly reduced liver TC and esterified cholesterol by up to 43%, and triglyceride levels by up to 60% (*p* < 0.01). Furthermore, corn bran oil also upregulated hepatic CYP7A1 involved in bile acid production by up to 49% (*p* < 0.01). As the dose of corn bran oil increased, the activity of hepatic acyl-CoA cholesterol acyltransferase (ACAT) involved in cholesterol storage decreased by up to 63% in a dose-dependent manner. Finally, corn bran oil intake led to an approximately 50% increase in fecal cholesterol excretion, suggesting corn bran oil promotes cholesterol elimination from the body. These findings suggest that corn bran oil has complex effects on cholesterol metabolism by affecting the lipoprotein synthesis and catabolism. It was later found that corn bran oil can reduce plasma cholesterol by changing hepatic cholesterol metabolism and upregulating LDL receptors, which are responsible for removing LDL cholesterol from the bloodstream [24,25]. In addition, previous research also proposed that the cholesterol-lowering effects of corn bran oil involved the disruption of bile acid enterohepatic circulation. The combined effect of these two mechanisms (cholesterol-lowering and bile acid absorption) led to a significant decrease in hepatic cholesterol concentrations, thereby upregulating LDL-LR and reducing plasma LDL-C concentrations. Furthermore, corn bran oil also increased the production of CYP7A1, an enzyme involved in bile acid production. This increase likely disrupted the normal absorption and circulation of bile acids in the gut, and the LDL receptor was upregulated, absorbing more cholesterol from the plasma, resulting in the observed decrease in plasma LDL cholesterol concentration.

Lime-treated corn husk (LTCH) is a by-product of the corn tortilla industry and offers an alternative use for this material. Studies compared the effects of LTCH on animal cholesterol levels to a diet rich in dietary fiber. The results showed that LTCH significantly reduced LDL-C and VLDL-C in the animal’s blood plasma. LTCH influenced enzymes involved in cholesterol metabolism by decreasing ACAT activity and increasing CYP7A1 activity. These results suggest that LTCH significantly reduces plasma cholesterol levels by inducing specific changes in the small intestine, thereby reducing hepatic microsomal cholesterol and altering hepatic cholesterol metabolism [105]. As a result of the changes, VLDL particles released into the bloodstream contain less cholesterol. These modified particles become less likely to be converted into LDL-C over time and might be cleared from the circulation faster by the body’s natural waste removal mechanism.

In general, the dietary fiber in corn bran can lower blood lipids through several mechanisms [83]. (1) Reduced cholesterol absorption. Corn bran fiber inhibits the activity of enzymes involved in cholesterol and lipid metabolisms [77]. This impedes the diffusion of cholesterol, thereby hindering cholesterol and triglycerides absorption. (2) Increased bile acid excretion. Corn bran fiber binds to bile acids preventing cholesterol reabsorption and promotes excretion through feces. Furthermore, a low amount of bile acid in the intestine promotes the conversion of cholesterol into bile acids, leading to lower blood cholesterol levels [106]. (3) Promoted short-chain fatty acid (SCFA) production and reduced cholesterol synthesis. Gut bacteria ferments dietary fiber to produce SCFAs, which influence cholesterol metabolism, potentially reducing cholesterol production while increasing its excretion [78,107]. (4) Improved cholesterol removal. Soluble dietary fiber binds to cholate, promoting intestinal movement, ultimately leading to increased cholesterol excretion [79,107]. Additionally, soluble fiber may also prolong the time that apolipoproteins stay in the bloodstream, allowing them to remove more cholesterol. As shown in Table 2.

### 2.5. Others (Corn Leaves, Corn Cobs)

Corn cob—the central stalk of a corn ear—has various traditional uses beyond its role in agriculture, such as to be used as a fuel source for drying and powering coal fires. In traditional Chinese medicine, it is used to strengthen spleen health and promote the elimination of dampness for conditions, such as difficulty in urination, edema, beriberi, summer heat in children, and sores on the mouth and tongue [108].

Recent research has explored the potential health benefits of various compounds extracted from the corn cob. One such compound, sweet corn cob polysaccharide (SCP-80-1), has shown promising in vitro hypoglycemic effects [109]. Its molecular weight and monosaccharide composition information are shown in Table 1. In previous studies, it has been proven to inhibit α-glucosidase [110,111], showing good in vitro hypoglycemic and anticoagulant effects. Studies in STZ-induced diabetic rats suggest that SCP-80-1 reduced both fasting and postprandial blood sugar levels, improved glucose tolerance, reduced organ swelling, and repaired pancreatic islet tissues. Diabetes is often linked to abnormal blood lipid levels, which can increase the CVD risk. Interestingly, SCP-80-1 appears to improve blood lipid profiles by reducing TC, TG, and LDL-C levels, while increasing HDL-C levels. These results suggest that SCP-80-1 helps regulate lipid metabolism, potentially reducing the risk of CVD in diabetic individuals [112,113]. The studies propose that SCP-80-I improves lipid metabolism by increasing HDL levels, transporting cholesterol from the bloodstream back to the liver for breakdown, and accelerating hepatic lipid degradation, thereby reducing blood lipid levels. As shown in Table 2.

Corn leaves—referring to the flat green structures on corn stalks—are essential for the plant’s growth and nutrient production. Traditionally, they have been used as livestock feed and fuel. However, recent research suggests that the corn leaves have blood sugar-lowering and anticancer properties [114]. Additionally, a study by Adeyi [115] found that the ethyl acetate extract of corn leaves (CLEA) helped reverse hematological disorders by normalizing the PCV, HB, WBC, and platelet counts. Compared with the diabetic nephropathy control group, the CLEA treatment group showed improvements in their urine tests, where the urinary total protein, albumin, glucose, and urea levels all returned to normal ranges after treatment. Therefore, the study suggests that corn leaf extract may be a potential alternative treatment for diabetic nephropathy [115]. The corn leaf extract may mitigate insulin resistance by inhibiting α-amylase and α-glucosidase, potentially leading to lower blood sugar levels after meals in diabetic patients and regulating blood lipids. However, more research is needed to confirm these findings and determine the safety and effectiveness. As shown in Table 2.

## 3. Conclusions and Perspectives

This review explores the exciting potential of corn wastes to help regulate blood lipid levels. This review suggests several ways that corn waste might achieve this, including (1) boosting oxidant capacity in the body, (2) inhibiting the absorption of dietary lipids, (3) promoting the excretion and efflux of bile acids, (4) increasing the activity and amount of HDL-C, (5) regulating insulin levels to prevent insulin resistance, and (6) regulating intestinal flora homeostasis, as shown in Figure 4. Furthermore, the presence of specific compounds in corn waste, such as flavonoids, phenols, and polysaccharides, that might play a role in these potential health benefits is also highlighted. While the effects are promising, more research is needed to unravel the mechanisms of action of how the phytochemicals and nutrients in the corn waste might influence blood lipid levels. It is crucial to pinpoint the specific bioactive substances within corn waste responsible for the therapeutic effects. By addressing these research gaps, the relevant stakeholders can unlock the potential of corn waste for developing new ways to manage dyslipidemia, eventually leading to the creation of safe and effective health products based on scientific discoveries. This review article highlights the potential of corn waste as a valuable resource for promoting health and sustainability. It encourages further research to explore its potential and contribute to the development of new approaches for managing dyslipidemia.

## Figures and Tables

**Figure 1 pharmaceuticals-17-00868-f001:**
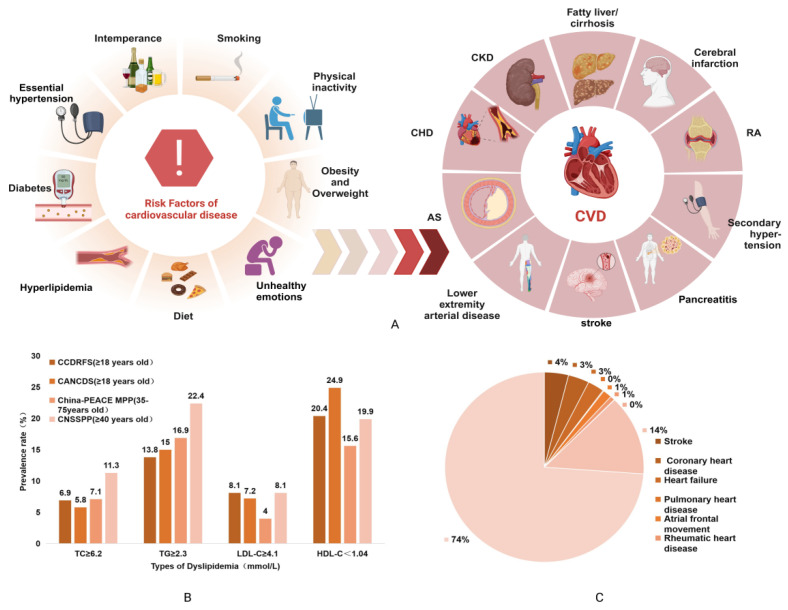
(**A**) Risk factors of CVD and its associated complications. (**B**) Comparison of the prevalence rates of different types of dyslipidemias based on the results of the fourth CCDRFS project in 2013–2014, the Chinese Adult American Cancer and Sexually Transmitted Disease Surveillance (CANCDS) project in 2015, the Chinese Prevalence Screening and Prevention Project (CNSSPP) in 2014, and the China-PEACE MPP project in 2014–2019 [6]. (**C**) The distribution of various *CVD* types in 330 million CVD cases in China [4].

**Figure 2 pharmaceuticals-17-00868-f002:**
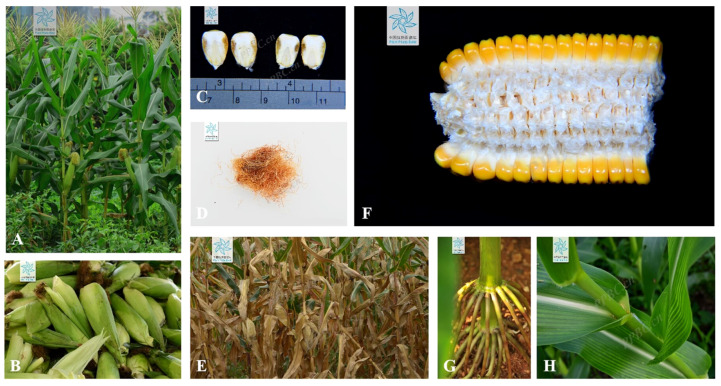
(**A**) Whole plant, (**B**) bracts, (**C**) brans, (**D**) silks, (**E**) stalks, (**F**) cobs, (**G**) roots, and (**H**) leaves of *Zea mays* L. [28].

**Figure 3 pharmaceuticals-17-00868-f003:**
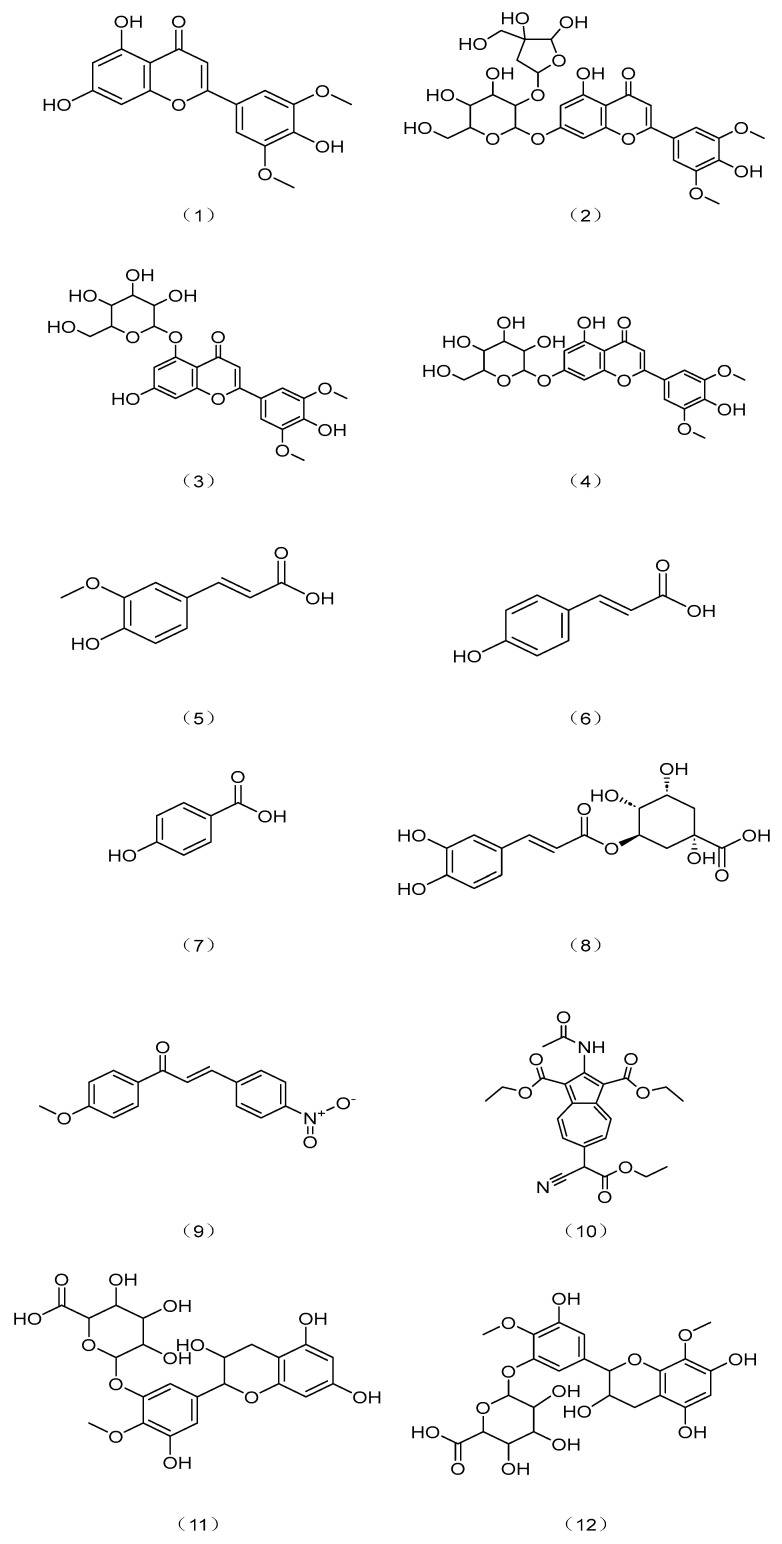
Structural formulas of compounds in corn wastes with blood lipid regulating functions.

**Figure 4 pharmaceuticals-17-00868-f004:**
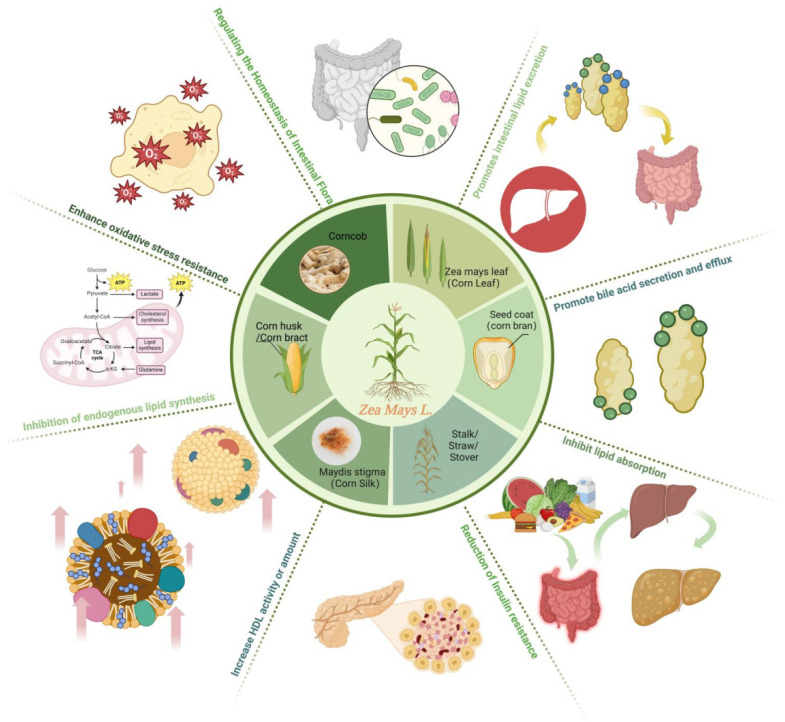
Corn (*Zea mays* L.) wastes and its mechanisms of regulating blood lipids [6].

**Table 1 pharmaceuticals-17-00868-t001:** Phytochemical composition of different corn wastes with blood lipid regulating function.

Plant Part	Class of Compound	Name of Compound	Chemical Formula	Molecular Weight	Structural Formula	Reference
corn bracts	Flavonoid	Tricin	C_17_H_14_O_7_	330.29	Compound (1) in Figure 3	[21]
corn bracts	Flavonoids	Tricin-5-O-β-D-glucoside	C_23_H_24_O_12_	492.4	Compound (2) in Figure 3	[29]
corn bracts	Flavonoids	Tricin-7-O-β-D-glucoside	C_23_H_24_O_12_	492.4	Compound (3) in Figure 3	[29]
corn bracts	Flavonoids	Tricin-7-O-[β-D-apifuranosyl(1→2)]-β-D-glucopyranoside	C_28_H_32_O_16_	——	Compound (4) in Figure 3	[29]
corn bracts, corn bran	phenolic acids	Ferulic acid	C_10_H_10_O_4_	194.18	Compound (5) in Figure 3	[22,30]
corn bracts	phenolic acids	p-coumaric acid	C_9_H_8_O_3_	164.16	Compound (6) in Figure 3	[22]
corn bracts	phenolic acids	p-hydroxybenzoic acid	C_7_H_6_O_3_	138.12	Compound (7) in Figure 3	[22]
corn bracts	phenolic acids	Chlorogenic acid	C_16_H_18_O_9_	354.31	Compound (8) in Figure 3	[22]
corn stalks	Flavonoids	trans-4′-methoxy-4-nitrochalcone	C_16_H_13_NO_4_	283.28	Compound (9) in Figure 3	[31]
corn stalks	Flavonoids	diethyl2-acetamido-6-(1-cyano-2-ethoxy-2-oxoethyl)-1,3-azulenedicarboxylate	C_23_H_24_N_2_O_7_	410	Compound (10) in Figure 3	[11]
corn stalks	Flavonoids	4′-methyl-epigallocatechin-3′-glucuronide	C_22_H_24_O_13_	496	Compound (11) in Figure 3	[11]
corn stalks	Flavonoids	4′,8-dimethoxy-epigallocatechin-3′-glucUronide	C_23_H_26_O_14_	526	Compound (12) in Figure 3	[11]
corn stalks	Phytosterols	sitosterol	C_29_H_50_O	414.7	——	[25]
corn stalks	Anthocyanin	anthocyanin	——	——	——	[27,32]
corn cob	Polysaccharides	SCP-80-1	Ara:Man:Glu:Gal= 0.369:0.824:10.759:0.333	18.350 kDa	——	[33,34]
corn silk	Polysaccharides	CSP-3	Man:Rha:Glu:Gal:Ara:Xyl:Gala= 1.53:0.00:0.43:1.06:0.46:0.35:1.46	5.9 ± 0.06 kDa	——	[35]

**Table 2 pharmaceuticals-17-00868-t002:** Possible active components with hypolipidemic effects, and the mechanisms by which different corn wastes might act against hyperlipidemia.

Name	Possible Active Components	Hypolipidemic Effects	Possible Mechanisms	References
corn bract	corn bract water extract	TC↓, TG↓, LDL-C↓, VLDL-C↓, HDL-C↑	Elevated HDL promotes RCT. Change the expression of p53, Fas, Bcl-2, and caspase-3 related factors of apoptosis and proliferation of VSMCs. Upregulate serum endothelin and prostacyclin in the VEC of atherosclerotic, regulate NO and ET levels, reduce endothelial apoptosis rate, promote endothelial repair, and resist high-fat injury. Increasing HDL inhibits the expression of adhesion molecules in endothelial cells, thereby reducing the recruitment of blood monocytes to the arterial wall.	[41,44,45,47,48,49,54]
corn bract	total flavonoids, total phenols	alpha-amylase↓, CAT↑, DPPH↑	Regulate α-amylase activity, reduce insulin resistance, inhibit lipolysis in adipose tissue, and stimulate the degradation of TG-rich lipoproteins in the circulation system. By protecting VEC, antioxidation, and removing excess oxygen free radicals in the body.	[22,59]
corn bract	tricin	TG↓, ALT↓, AST↓	Reduce lipids by inhibiting the biosynthetic pathway of TC\TG in vivo by regulating the AMPK signaling pathway. Inhibit HMGCR, SREBP, FAS, ACC, and CPT-1 and reduce TG synthesis.	[60,61]
cornstalk	flavonoids	alpha-amylase↓	Inhibiting α-amylase increases insulin content, reduces postprandial hyperglycemia in diabetic patients, and regulates blood lipid levels. Improve the scavenging capacity of DPPH, ABTS+, ·OH, and the total reduction capacity of Fe^3+^ to regulate lipid metabolism.	[11,31]
corn silk	corn silk water extract	TC↓, TG↓, LDL↓, HDL↑, SOD↑, MDA↓, GSH-PX↑, ALT↓, AST↓	Prevent cholesterol synthesis in the liver. Promote the breakdown of cholesterol and fatty acids in the body. Antioxidative stress and protection of liver tissue. Increase HDL and enhance reverse cholesterol transport to lower blood lipids. Activates the AMPK signaling pathway and promotes the decomposition of cholesterol and fatty acids in the body.	[64,65,66]
corn silk	corn silk polysaccharide	TC↓, TG↓, LDL-C↓, HDL-C↑	Increase HDL and strengthen the reverse transport of cholesterol and apparent blood lipids in blood vessels.	[35,67,68]
corn silk	corn silk acidic polysaccharide	TC↓, TG↓, Kch↓, LDL-C↓	Increase HDL and strengthen the reverse transport of cholesterol and apparent blood lipids in blood vessels.	[69]
corn silk	flavonoids	TC↓, TG↓, LDL-C↓, HDL-C↑, SOD↑, MDA↓	It may synergistically lower lipids by inhibiting endogenous synthesis, exogenous absorption, and cholesterol efflux.	[70]
corn bran	dietary fiber	TC↓, TG↓	Reduce the activity of enzymes related to lipid metabolism and reduce the absorption of cholesterol. By adsorbing bile acids, it promotes the conversion of cholesterol. Promote the production of short-chain fatty acids and inhibit cholesterol synthesis. Bind cholate and promote cholesterol excretion.	[71,72]
corn bran/corn fiber oil	phytosterols	TC↓, TG↓, LDL-C↓	Lowering cholesterol by reducing cholesterol absorption and increasing bile acid output. Alterations in hepatic cholesterol metabolism and upregulation of LDL receptors reduce plasma cholesterol. Reduce intestinal cholesterol absorption through competitive inhibition.	[73,74,75]
corn bran	lime-treated corn brans	LDL-C, VLDL-C↓	Downregulation of ACAT activity and upregulation of CYP7A1 activity significantly reduce plasma cholesterol levels by inducing specific changes in the small intestine and removing or reducing the rate of conversion to LDL.	[76]
corn cob	polysaccharide (SCP-80-1)	alpha-amylase↑, TC↓, TG↓, LDL-C↓, HDL-C↑	It may be that SCP-80-I reduces blood lipid levels by increasing serum HDL levels, transporting cholesterol to the liver, accelerating liver lipid degradation, and thus reducing blood lipid levels.	[77]
corn leaves	ethyl acetate extract	-	By inhibiting α-amylase and α-glucosidase, insulin levels can be increased, and postprandial hyperglycemia in diabetic patients can be reduced. Regulating blood lipids requires further verification.	[78,79]

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
