# Peer review of "Research on the Mechanism and Material Basis of Corn (Zea mays L.) Waste Regulating Dyslipidemia"

_pharmaceuticals, 2024, doi:10.3390/ph17070868_

Round 1
Reviewer 1 Report
Comments and Suggestions for Authors
This manuscript presents an interesting review of the use of corn waste for medicinal purposes. I believe this will be a valuable contribution to the field that will be informative for researchers wishing to know more about the medicinal properties of corn waste. However, there are a number of minor issues that should be corrected before the manuscript is published:
Line 59: Atherosclerosis abbreviation (AS) should be defined here and used consistently throughout the manuscript.
Line 61: This line is unclear. These factors increase the probability and severity of tumors in the body?
Figure 1: some of the text in the figure is a bit small and not very visible. Please enlarge.
Line 90: Keep consistency in the list, both in comma use and in wording: e.g. “regulate insulin levels” should be “regulating insulin levels” to match previous items in the list.
Line 108: “They can be used” rephrase: either “They have also been used” or “They have the potential to treat inflammation” instead.
Line 111: use distinct formatting for subheadings so they stand out from the body text.
Line 125: “preventing insulin concentrations” is unclear. Does this mean lowering insulin concentrations? Or increasing insulin concentrations?
Line 140 refers to “studies” but only cites one paper
Line 170, 171: “As” should be “AS”
Line 255: alfalfa reference is unclear.
Line 276: “Metabolism Mainly includes” is unclear. Do you mean “AMPK mainly regulates metabolism through:”
Line 280: What is “lip synthase”? ACC I assume refers to Acetyl Co-A Carboxylase, which regulates fatty acid synthesis
Table 1: use subscripts for numbers in chemical formulas
Adjust heading so that chemical formula is on two lines and molecular weight isn’t split into so many lines
Include units in molecular weight heading (g/mol, which is equal to Da)
Line 293: “swelling-reducing”
Line 297: “in the form of boiled water” why not say “decoction”?
Line 349: Explain further. Hypoglycemic conditions could also potentially increase blood lipid levels due to increased TAG breakdown
Line 361 needs to be supported by citations
Line 369: “played at least an important role” is unclear. Played an important role in what?
Line 372: contains extra period
Line 375 needs to be supported by citations
Line 376 transition is weak. Try something like this “In addition to flavonoids…”
Line 378: would be better as “many studies show a high content of beta-sitosterol in corn silk”
Line 411 needs to be supported by citations
Line 453 is unclear. Corn silk was included in their feed? Or it was applied as a water extract?
Line 454. TC and TG were collected by water extraction?
Line 465 refers to “studies” but only cites one paper
In figure 3, is it indicating that all corn waste has all those effects? Figure could be better if different products were linked clearly to different effects.
Line 528: these two sentences should be condensed and combined
Line 538 sentence intro seems to be missing. What significantly lowers plasma cholesterol levels?
Line 608 “discovery of diabetes”? progression or development would be a better word choice here
Line 621. Use consistency in how you refer to authors (i.e. full first name and last name used in other places)
Line 629: “Content” does this refer to the content of corn leaves in patients?
Table 2: mechanisms section is way too long and ruins the formatting of the remainder of the table. I would shorten this in the table and expand on it in the text instead.
Abbreviations: TC is commonly used as an abbreviation for total cholesterol. This should be specified in the table. Abbreviations are used inconsistently in the text. Some abbreviations are also defined in the text, and other abbreviations are not used in some places but not throughout the text. Abbreviations should either all be defined in the text or none should be. Furthermore, abbreviations should be consistently used each time the term is used.
Comments on the Quality of English LanguageThe quality of English language is not great. I have included some of my suggestions in my previous suggestions, but I would suggest going over the whole manuscript for other English errors as well.
Reviewer 2 Report
Comments and Suggestions for Authors
Grammatic and stylist revision are mandatory
In the last sentence of Introduction, 'et al. They contain' after [27] should be deleted.
The paper lacks appropriate references in the following sections, 2.1.2, 2.1.2.3, 2.1.2.4, 2.2.2, 2.3, 2.3.1, 2.3.2.`, 2.4.1
;A study by Miura (2.3.2.3) should be referenced
Comments on the Quality of English LanguageLanguage presentation is below average. Professional and expert language revision is mandatory. Literature need to be updated
Reviewer 3 Report
Comments and Suggestions for Authors
Reviewer 4 Report
Comments and Suggestions for Authors
This manuscript describes the corn waste products that can regulate blood lipids. It also provides the mechanisms of its action with its chemical composition.
However, some points need to be corrected, especially those related to the language. Please find the attached pdf file.
I think the biggest issue in this manuscript is that there are many references in the Chinese local journals and the Chinese language, and it is difficult to find or download them. I do not know if there is a limited number of references in languages other than English to be used in Pharmaceuticals.

Comments on the Quality of English LanguageThis manuscript describes the corn waste products that can regulate blood lipids. It also provides the mechanisms of its action with its chemical composition.
However, some points need to be corrected, especially those related to the language. Please find the attached pdf file.
I think the biggest issue in this manuscript is that there are many references in the Chinese local journals and the Chinese language, and it is difficult to find or download them. I do not know if there is a limited number of references in languages other than English to be used in Pharmaceuticals.
Reviewer 5 Report
Comments and Suggestions for Authors
Although it is an interesting review, it has some details that do not allow it to be accepted for publication in the current form.
Table 1 . Points 5,6 and 7, are they phenols or phenolic acids?
Please improve the tables for example in the row of column headings: molecular weight, structural formula, references, ideally, they should be in one line. I don't know if it is possible to make the table horizontally.
chemical composition or phytochemical composition? the document states flavonoids, phenolic acids, alkaloids, sterols, and dietary fiber, sterols, depending on the part of the maize used... not in general.
2.1.2 The Mechanism of Corn Husk in Lowering Blood Lipids please add Reference?
Please check Table 2., especially in the column of names. align it to correspond with the other columns.
2.1.2.1 Increase HDL-C activity and quantity, in this section it mentions several studies, but only mentions one (33).
Line 181. In addition, the elevated HDL in corn bracts can... it is correct?
Line 245. corn bract remove DPPH, and whether it removes it or only inhibits its action?
Line 255. Alfalfa???
2.4.2 The Mechanism of Corn Bran in Lowering Blood Lipids.
In this paragraph please check the spaces as well as the use of capital letters, spaces between words, etc.
Check typos throughout the document.
Improve the tables
Round 2
Reviewer 5 Report
Comments and Suggestions for Authors
The authors have responded adequately to the doubts and comments made on their manuscript.
two minor details
In the scientific name of maize L. it is not italicized since it is derived from Linneus.
Table 2. In the first column, please start with a capital letter.
